# Interferon-Inducible Protein 10 and Disease Activity in Systemic Lupus Erythematosus and Lupus Nephritis: A Systematic Review and Meta-Analysis

**DOI:** 10.3390/ijms20194954

**Published:** 2019-10-08

**Authors:** Pongpratch Puapatanakul, Sonchai Chansritrakul, Paweena Susantitaphong, Thornthun Ueaphongsukkit, Somchai Eiam-Ong, Kearkiat Praditpornsilpa, Wonngarm Kittanamongkolchai, Yingyos Avihingsanon

**Affiliations:** 1Division of Nephrology, Department of Medicine, Faculty of Medicine, Chulalongkorn University, Bangkok 10330, Thailand; pongpratpk@gmail.com (P.P.); pesancerinus@hotmail.com (P.S.); somchai80754@yahoo.com (S.E.-O.); kearkiat@hotmail.com (K.P.); wonngarm.k@gmail.com (W.K.); 2Department of Medicine, Chonburi Hospital, Chonburi 20000, Thailand; sonchai07@yahoo.com; 3Department of Medicine, Faculty of Medicine, Chulalongkorn University, Bangkok 10330, Thailand; thornthun.uea@gmail.com; 4Renal Immunology and Transplantation Research Unit, Faculty of Medicine, Chulalongkorn University, Bangkok 10330, Thailand; 5Center of Excellence in Immunology and Immune-mediated Diseases, Faculty of Medicine, Chulalongkorn University, Bangkok 10330, Thailand

**Keywords:** systemic lupus erythematosus, lupus nephritis, interferon-inducible protein 10, IP-10, CXCR10

## Abstract

There is increasing evidence of a correlation between interferon-inducible protein 10 (IP-10) and disease activity of systemic lupus erythematosus (SLE) and lupus nephritis (LN). We conducted a comprehensive search on IP-10 using MEDLINE, Scopus, and Cochrane electronic databases from the beginning to the end of December 2017. All studies that compared serum and/or urine IP-10 between active SLE/LN patients and any control groups were identified and included in this systematic review and meta-analysis. The mean difference (MD) of IP-10 level among active SLE and LN patients, as well as the correlation of IP-10 with disease activity, were meta-analyzed using a random-effects model. From 23 eligible studies, 15 provided adequate data for meta-analysis. Serum IP-10 was significantly elevated in patients with active SLE compared to non-active SLE patients (MD 356.5 pg/mL, 95% CI 59.6 to 653.4, *p* = 0.019). On the other hand, the levels of serum IP-10 was not different between active LN and non-active LN. However, serum IP-10 was positively correlated with disease activity like SLE disease activity index (SLEDAI) (pooled *r* = 0.29, 95% CI 0.22 to 0.35, *p* < 0.001). Furthermore, urine IP-10 tended to be higher in patients with active LN compared to non-active LN patients but this did not reach statistical significance (MD 3.47 pg/mgCr × 100, 95% CI −0.18 to 7.12, *p* = 0.06). Nevertheless, urine IP-10 was positively correlated with renal SLEDAI (pooled r = 0.29, 95% CI 0.05 to 0.50, *p* = 0.019). In conclusion, serum and urine IP-10 levels may be useful in monitoring the disease activity of SLE and LN. Serum IP-10 was correlated with systemic disease whereas urine IP-10 was a useful biomarker for detecting active LN.

## 1. Introduction

Despite the availability of new treatments directed against causative molecular targets, systemic lupus erythematosus (SLE) patients still suffer from disease flares due to the unpredictable nature of the disease. Conventional biomarkers are suboptimal in detecting flares and its sensitivity is 50% and its specificity is 75% [1,2]. Therefore, we are in need of better biomarkers that can accurately inform us about the activity of the disease.

It has been established that both innate and adaptive immune responses were involved in SLE pathogenesis. Throughout the years, evidence from multiple studies showed that interferon (IFN) signaling pathway had a pivotal role in SLE. Early studies showed a high level of IFN in the serum of patients with SLE [3,4]. The use of IFN as a treatment in various diseases could also induce autoantibodies seen in SLE [5]. More recently, gene expression studies showed that there was an overexpression of IFN-stimulated genes (ISGs) in the blood of SLE patients, which highlighted the significant role of IFN in SLE [6,7]. Further study on ISG products has identified many promising IFN-regulated chemokines and demonstrated the correlation of these chemokines with disease activity making them possible biomarker candidates [8].

IFN gamma inducible protein 10 (IP-10) or chemokine ligand 10 (CXCL10) is of particular interest because it is one of the IFN-regulated chemokines that has the strongest correlation with SLE disease activity as previously reported [8]. It is a member of T-helper 1 lymphocyte chemokines expressed and secreted by the infiltrating monocytes, macrophages, and endothelial cells in response to IFN stimulation. These cells then bind to the receptor CXCR3, which is expressed on T-lymphocytes, and is responsible for T-lymphocyte trafficking into the affected organs of SLE patients. This is evident in the murine lupus model showing an increased expression of IP-10 and its receptor CXCR3 correlating with the migration of lymphocytes into lung tissue [9]. In patients with lupus nephritis (LN), the increased expression of CXCR3 correlated with worse renal function, a higher degree of proteinuria, and percentage of globally sclerosed glomeruli [10,11]. Repeated renal biopsy in 113 SLE patients after induction therapy for active LN also showed that tubulointerstitial IP-10 expression significantly decreased when proliferative or mixed nephritis changed to membranous nephropathy [12].

Levels of serum and urine IP-10 have been investigated in patients with SLE and lupus nephritis (LN) in various clinical settings [13,14,15,16]. Serum IP-10 level was significantly higher in SLE patients compared to healthy age- and sex-matched healthy controls or in patients with rheumatoid arthritis [17]. Urinary messenger RNA level of IP-10 was very specific for detecting diffuse proliferative LN [13] and serial measurement of serum or urine levels of IP-10 was helpful in predicting SLE flare and LN, respectively [13,18]. However, there were some conflicting results regarding the utility of serum and urine IP-10 in SLE and LN [19,20]. Therefore, we conducted a systematic review and meta-analysis of all studies that used serum or urine IP-10 in SLE patients with or without LN.

## 2. Results

### 2.1. Study Characteristics

Applying the proposed search strategy, a total of 297 publications were identified. Duplicated publications and titles or abstracts that did not have the keywords we wanted were excluded from this systematic meta-analysis. After eliminating such publications, we were left with 33 publications. We evaluated these 33 publications in detail. We found that 23 publications fulfilled the eligibility criteria (3006 patients; Figure 1) [13,14,15,16,17,18,19,20,21,22,23,24,25,26,27,28,29,30,31,32,33,34,35]. While all studies investigated SLE patients, only 6 of them focused on LN [13,19,22,25,27,31]. Fifteen studies had available data for additional meta-analysis (1921 patients) [14,15,16,17,18,19,21,24,25,26,27,28,31,32,35]. The detailed characteristics of the selected studies are shown in Table 1.

The majority of the studies were conducted in Asia (11 studies), while others were from Europe, northern America, and Africa (eight, two, and two studies, respectively). There were 15 cross-sectional studies [14,17,19,21,22,23,24,26,27,29,31,32,33,34,35] and eight prospective cohort studies [13,15,16,18,20,25,28,30]. All studies were conducted in adults except for one study which was done in children with juvenile-onset SLE [19]. Disease activity was determined by the clinical score in the majority of the studies while only one study used renal pathology grading according to the International Society of Nephrology/Renal Pathology Society 2003 classification for LN [36]. The most commonly used disease activity score was SLEDAI or revised versions of SLEDAI (SELENA, SLEDAI and mSLEDAI-2K), which were used in 16 studies. Otherwise, the British Isles Lupus Assessment Group (BILAG) index, Systemic Lupus Activity Measure (SLAM) index, and Physicians’ Global Assessment (PGA) were used in three, two, and one study, respectively, while four studies did not clearly report the criteria for active disease. IP-10 was measured in the serum, urine, or both in 17, three, and three studies, respectively. 21 studies measured the concentration of IP-10 protein by using enzyme-linked immunosorbent assay (ELISA) (17 studies), cytometric bead array (3 studies), or multiplex immunoassay (1 study), while the other two studies measured the mRNA transcript of IP-10 through quantitative reverse transcription-polymerase chain reaction (RT-PCR) (Table 1). Multiple pairs of comparisons were available. Ten studies compared active SLE with inactive SLE [15,16,17,19,20,21,25,28,30,31] while 19 studies provided comparison between active SLE with healthy controls or other groups that were non-SLE patients [13,14,16,17,19,20,21,22,23,24,26,27,29,30,31,32,33,34,35]. Patients with active LN were compared with either active SLE without renal involvement (eight studies) [14,17,20,25,27,28,31,32], inactive SLE (nine studies) [13,14,17,19,20,21,25,28,31], non-SLE controls (10 studies) [13,14,17,19,20,21,22,28,31,32], or LN of other classes (two studies) [13,22]. The unit of measurement for serum IP-10 concentration was uniform in pg/mL except for two studies, which reported the level of serum IP-10 in ng/mL [29] or chemiluminescence intensity [33]. On the other hand, the measurement units for urine IP-10 were diversified including pg/mgCr, pg/mgCr x 100, mg/gCr × 10^5^, pg/mL, and pg/dL. The IP-10 mRNA transcripts were reported in copies/µg of total RNA or fold-change.

### 2.2. Quality Assessment

According to the QUADAS-2 tool, the unclear risk for bias was remarkable regarding the index test (Figure 2). This was mainly due to the fact that almost all of the included studies measured the IP-10 without blinding the participants’ statuses. The interpretation of IP-10 was not stated in any of the studies which raised the question whether the interpreters knew the results of the reference standards or not. Anyway, the IP-10 results were objective which likely lessened the impact of such bias on the study results. There was a risk of selection bias of the participants and the definition criteria of active disease were not clearly stated. There was a minimal risk of bias that was detected in the study flow and timing. Nonetheless, the applicability of patient selection, index test, and reference standard were of no concern.

### 2.3. Serum IP-10 and SLE

Data from all studies investigating serum IP-10 showed that serum IP-10 from active SLE patients was significantly higher than the healthy controls. The mean level of serum IP-10 in SLE patients varied among studies, ranging from 73.1 to 511.0 pg/mL if measured by the conventional ELISA and could be as high as 2,030.0 to 7,121.4 pg/mL when measured by cytometric bead array or multiplex immunoassay [14,35]. The meta-analysis of eight studies [16,17,21,24,26,31,32,35] with available numeric data for this comparison (*n* = 1069 patients, 769 active SLE patients, 300 healthy controls) revealed that serum IP-10 in active SLE patients was significantly higher than the healthy controls (mean difference [MD] 153.9 pg/mL, 95% confidence interval [CI] 91.6 to 216.1, *p* < 0.001). There was high heterogeneity between the studies as evidenced by *I*^2^ index of 87.1% (*p* < 0.001) and had a potential publication bias (*p* = 0.04) (Table 2).

The difference between serum IP-10 of active and inactive SLE patients was conflicting. Six studies observed that serum IP-10 in active SLE patients was significantly higher than inactive patients [15,16,17,25,28,31]. One study showed significantly higher serum IP-10 in patients with severe SLE compared to those with moderate disease activity [16], while three studies observed insignificant differences [20,21,30]. However, in the meta-analysis of five study arms from four studies with available data (*n* = 897 patients, 122 active SLE patients, 775 inactive SLE patients) [16,17,25,31], serum IP-10 in active SLE patients was significantly higher than inactive patients (MD 356.5 pg/mL, 95% CI 59.6 to 653.4, *p* = 0.019). There was high heterogeneity between studies as evidenced by an *I*^2^ index of 95.9% (*p* < 0.001) (Table 2).

Nine studies investigated the correlation between serum IP-10 and disease activity index. Seven [14,15,24,25,28,31,32] and two reports [18,28] correlated serum IP-10 with the SLEDAI and BILAG indices, respectively. The pooled correlation analysis revealed that serum IP-10 was positively correlated with SLEDAI (pooled correlation coefficient [*r*] = 0.29, 95% CI 0.22 to 0.35, *p* < 0.001) and BILAG index (pooled *r* = 0.41, 95% CI 0.24 to 0.56, *p* < 0.001). There was no heterogeneity as the *I*^2^ index was 0% with *p* = 0.61 and 0.55, respectively, in both analyses (Table 3).

Serum IP-10 was also correlated with other biomarkers. Five study arms in three studies correlated serum IP-10 with complement C3 and C4 levels (*n* = 1,096) and was meta-analyzed showing significant negative correlation (pooled *r* = −0.20, 95% CI −0.30 to −0.10, *p* < 0.001) [16,17,32]. Correlation between serum IP-10 and anti-dsDNA and erythrocyte sedimentation rate from the same reports was also analyzed showing a significant positive correlation (pooled *r* = 0.28, 95% CI 0.15 to 0.40, *p* < 0.001). There was moderate heterogeneity of studies in the latter correlation (*I*^2^ index = 70.3%, *p* = 0.01) (Table 3).

In two studies, serum IP-10 was correlated with SLE-related hematologic abnormalities. Serum IP-10 negatively correlated with the number of white blood cells (*r* = −0.423), polymorphonuclear cells (*r* = −0.303), lymphocytes (*r* = −0.386), and monocytes (*r* = −0.365) in one study [26] and in another study, serum IP-10 was negatively correlated with hemoglobin (*r* = −0.315) and total white blood cell count (*r* = −0.272) [28]. One study focused on pulmonary involvement in SLE and observed that serum IP-10 negatively correlated with total lung capacity (*r* = −0.59) and positively correlated with airway resistance (*r* = 0.55) [33].

There were two studies that compared the presence of serum IP-10 in patients with SLE and non-SLE patients with other connective tissue diseases (19 rheumatoid arthritis, 21 systemic sclerosis, and 28 polymyositis/dermatomyositis). The serum IP-10 in SLE patients was significantly higher than rheumatoid arthritis patients [17] but was not significantly higher than systemic sclerosis and polymyositis/dermatomyositis patients [29].

Six studies provided ROC analysis for serum IP-10. One study showed that the serum IP-10 appeared to be a good biomarker for detecting active SLE with the area under the ROC curve of 0.77 (95% CI 0.68−0.84), which was not better than complement C3, C4, and anti-dsDNA [25]. However, in another study, serum IP-10 outperformed anti-dsDNA [16]. Serum IP-10 also performed fairly well at predicting flare in two studies (area under the ROC curve ranging from 0.648 to 0.75) [18,28]. Moreover, in one study, it was shown that serum IP-10 was a good biomarker for detecting pulmonary involvement (area under the ROC curve 0.815, 95% CI 0.565 to 0.955) [33].

### 2.4. Serum IP-10 and LN

Three studies that reported a comparison of serum IP-10 between LN and healthy controls were included in the meta-analysis (*n* = 193 patients, 82 LN patients, and 111 healthy controls) [14,31,32]. Serum IP-10 was significantly higher in LN patients (MD 183.8 pg/mL, 95% CI 126.5 to 241.1, *p* < 0.001). There was no heterogeneity among the studies (*I*^2^ index = 0%, *p* = 0.37) (Table 2).

To differentiate between SLE patients with active LN from SLE patients without LN, there were seven study arms from five studies that provided adequate data for meta-analysis (*n* = 402 patients, active LN 201 patients, non-LN SLE 201 patients) [14,25,28,31,32]. Serum IP-10 of SLE patients with active LN was not significantly different from those without LN (MD 22.6 pg/mL, 95% CI −182.8 to 228.1, *p* = 0.83). There was high heterogeneity among the five studies (*I*^2^ index = 86.9%, *p* < 0.001) (Table 2). Two studies provided ROC analysis of serum IP-10 in detecting renal involvement in SLE; these studies showed that serum IP-10 was a poor biomarker in detecting renal involvement in patients with SLE (area under the ROC curve ranging from 0.595 to 0.633) [31,32].

### 2.5. Urine IP-10 and SLE

Five studies investigated the level of urine IP-10 in SLE patients with and without active LN [19,25,27,31,32], three of which also had data from healthy controls [19,31,32]. Since there were different measurement units among the studies using the ELISA technique, the measurement units were converted into pg/mgCr x 100 in three out of five studies. The other two studies reported urine IP-10 in pg/mL or pg/dL, which were not normalized by urine creatinine and thus could not be properly converted [27,32]. Interestingly, meta-analysis of the two studies (*n* = 95 patients, 62 active SLE patients, 33 healthy controls) showed that there was no significant difference between urine IP-10 of SLE patients without LN and healthy controls (MD 0.21 pg/mgCr x 100, 95% CI −0.74 to 1.15, *p* = 0.67) [19,31]. There was no heterogeneity for this comparison (*I*^2^ index = 54.3%, *p* = 0.14). Moreover, comparison of urine IP-10 between active and inactive SLE without LN (*n* = 156 patients, 88 active SLE patients, 68 inactive SLE patients) also showed that there were no significant differences between both groups (MD 2.81 pg/mgCr x 100, 95% CI −2.40 to 8.01, *p* = 0.29). There was a high heterogeneity between these studies (*I*^2^ index = 90.1%, *p* = 0.001) (Table 2).

Out of the five studies mentioned in the above paragraph, four studies correlated urine IP-10 with disease activity index for SLE, three of which could be meta-analyzed (*n* = 236 patients) [25,31,32]. Urine IP-10 showed significant positive correlation with SLEDAI (pooled *r* = 0.21, 95% CI 0.05 to 0.36, *p* = 0.011). There was no heterogeneity among the studies (*I*^2^ index = 27.4%, *p* = 0.25) (Table 3).

### 2.6. Urine IP-10 and LN

Six studies investigated urine IP-10 in LN patients. Five out of six studies provided comparison of urine IP-10 between active LN and active SLE without LN patients or inactive SLE patients [19,25,27,31,32]. Due to different measurement techniques and units used in these studies, only five study arms comparing LN with either active SLE without LN or inactive SLE from three studies could be meta-analyzed [19,25,31]. There was a trend of higher urine IP-10 in LN but this did not reach statistical significance (MD 3.47 pg/mgCr x 100, 95% CI −0.18 to 7.12, *p* = 0.06). There was a high heterogeneity among these three studies (*I*^2^ index = 88.2%, *p* < 0.001) (Table 2). However, in one study, urine IP-10 showed significant positive correlation with a degree of proteinuria (*r* = 0.72) and renal pathology class of LN II to IV (*r* = 0.84) [27]. Three studies reported positive correlation between urine IP-10 and renal SLEDAI (pooled *r* = 0.29, 95% CI 0.05 to 0.50, *p* = 0.019) [25,27,32]. There was moderate heterogeneity (*I*^2^ index = 73.8%, *p* = 0.02) (Table 3).

One study compared urine mRNA transcript of IP-10 among different classes of LN showing that class IV LN, according to ISN/RPS classification, had significantly higher urine mRNA for IP-10 compared to non-class IV LN (classes II, III, V, and VI) [13].

Five studies reported ROC curve analysis to demonstrate the overall performance of urine IP-10 on differentiating LN in SLE patients. In one study, urine mRNA transcript of IP-10 performed better than the conventional biomarkers such as 24-h urine protein, creatinine clearance, and urinary sediments in distinguishing class IV LN from other classes of LN with the area under the ROC curve of 0.89 (95% CI 0.78 to 0.99), which had a 73% sensitivity and 94% specificity at the cut-off of 2.09 log copies/µg total RNA [13]. In three studies, urine IP-10 appeared fair at distinguishing LN from SLE patients without LN with the area under the ROC curve ranging from 0.595 to 0.68 which could not outperform those of conventional biomarkers including complement C3 level or anti-dsDNA antibody [25,31,32] while another study showed that urine IP-10 was an excellent tool that could detect LN (area under the ROC curve 1.000) [27]. One study also showed that urine IP-10 was good at distinguishing active SLE with LN from active SLE without LN (area under the ROC curve 0.700, 95% CI 0.431 to 0.969) [31].

### 2.7. Serum IP-10 vs urine IP-10

Three studies measured both serum and urine IP-10 in each patient (*n* = 307 patients, 80 active non-LN patients, 84 active LN patients, 72 inactive SLE patients, and 71 healthy controls) [25,31,32]. Although the data could not be meta-analyzed, all three studies provided the area under the ROC curve analysis. Only one study by El-gohary et al. analyzed both serum and urine IP-10 in every patients and showed that serum IP-10 was better than urine IP-10 in differentiating active from inactive SLE (area under the ROC curve 0.753, 95% CI 0.594 to 0.911 vs 0.654, 95% CI 0.467 to 0.841, respectively) [31]. Another study by Abujam et al. only analyzed serum IP-10 for SLE and urine IP-10 for LN and hence did not provide a direct comparison of performance between serum and urine IP-10 [25]. The other study by Choe et al. focused mainly on LN and found that serum and urine IP-10 performed comparably in detecting SLE with LN (area under the ROC curve 0.595, 95% CI 0.46 to 0.73 vs 0.519, 95% CI 0.38 to 0.66, respectively) [32].

## 3. Discussion

In this systematic review and meta-analysis, serum IP-10 levels were significantly elevated during the flare of SLE patients. Serum IP-10 levels were associated with disease activity indices such as SLEDAI or BILAG. Therefore, serum IP-10 should be a marker for detecting active disease in SLE. However, there were no changes in the level of serum IP-10 during active LN. Serum IP-10 was associated with SLE-related hematologic or pulmonary involvements but there were few studies available for meta-analysis. Serum IP-10 levels are in accordance with serum monocyte chemoattractant protein-1 (MCP-1). Both are interferon-inducible chemokines and their serum levels were associated with systemic disease activity. Also, urine MCP-1 may be used for detecting disease activity in LN [37].

In this meta-analysis, the urine IP-10 level was not significantly elevated in active LN patients. The performance of urine IP-10 by area under the ROC curve in detecting LN from a study utilizing both serum and urine specimens was also unsatisfactory. However, publications on urine IP-10 had a high heterogeneity. Methods to measure urine IP-10, the unit of measurement and normalization method were so different between the studies making it difficult to compare the effects of urine IP-10. In fact, urine IP-10 levels had a trend to increase in active LN. Three out of six studies supported the fact that levels of urine IP-10 could distinguish active LN from others. On the other hand, two studies by Watson et al. and El-Gohary et al. found urine IP-10 levels were not significantly elevated in active LN patients [19,31]. Both studies used the ELISA kit bought from the R&D system and reported urine IP-10 normalized by urine creatinine levels (pg IP-10 per mg Cr). Because of this, it is difficult to standardize and compare the value of urine IP-10 between the studies when different measurement units are used. Other factors that have been shown to vary much between the studies are the concentration of the urine, amount/volume of the urine, interference of urine cells and heterogeneity of the study populations. All of these factors can contribute to conflicting results pertaining to urine IP-10′s function in LN. It remains inconclusive whether urine IP-10 protein can be used as a biomarker to predict active LN.

Three studies demonstrated that there was a positive correlation between serum IP-10 and the conventional serum biomarker, anti-dsDNA antibody, among SLE patients [16,17,32]. One study even showed a significant correlation during serial measurements [17]. This significant correlation confirms their connection in the pathogenesis of SLE. Basically, anti-dsDNA antibody can form immune complexes and activate IFN signaling pathway which ultimately leads to IP-10 production [38,39]. Anyway, anti-dsDNA antibody is still far from being an ideal biomarker for predicting SLE disease activity. Persistent elevation of anti-dsDNA antibody in patients with no evidence of active disease or normal anti-dsDNA antibody levels in patients with obvious disease activity have been reported [40]. This infers that the inducer of IFN signaling pathway in SLE is not limited to anti-dsDNA antibody. IP-10, on the other hand, represents the downstream mediators of the IFN-regulated pathways and may better correlate with disease activity. However, studies comparing area under the ROC curve analysis between IP-10 and anti-dsDNA antibody are limited and yielded conflicting results [16,18,25].

A major limitation of these studies was that there was little information from the kidney biopsy which is the gold standard for detecting SLE and LN. Also, the data was limited so we could not ascertain the best cutoff value for IP-10 to predict when the disease will flare. To overcome this, measurement of IP-10 mRNA transcript by quantitative RT-PCR, which was used by two studies in our systematic review showing highest urine IP-10 expression among patients with class IV LN (the most severe class), is another promising alternative technique for IP-10 measurement.

## 4. Materials and Methods

### 4.1. Literature Search

We conducted a systematic search and retrieval of literature from online sources, such as MEDLINE, Scopus, and Cochrane electronic databases from the beginning through to the end of December 2017. The search terms used were ′lupus nephritis′ or ′systemic lupus erythematosus′ combined with ′interferon-inducible protein 10′, ′IP-10′, or ′CXCL10′, all of which were used as medical subject heading terms or keywords for each respective database. To complete the literature search, we also screened references of relevant studies cited in these publications.

### 4.2. Study Selection Criteria

Only the studies fulfilling the following inclusion criteria were included: (1) the studies were prospective cohort, case-control, or of cross-sectional design; (2) serum or urine IP-10 was investigated to detect disease activity in SLE patients with or without LN; (3) there were comparisons between at least 2 groups (active SLE patients, inactive SLE patients, non-SLE patients and/or healthy controls); (4) the patients met the classification for SLE as per the 1997 updated American College of Rheumatology (ACR) criteria [41,42]. If there were more than one publication using the same set of data, we utilized the data from the most relevant and conclusive report. There were no language or sample size restrictions. Titles and abstracts of all articles were screened by two reviewers (P.P. and S.C.). The studies based on animal or cell cultures, and conference abstracts without subsequent publication in full-text were excluded. All discrepancies were reconciled by consensus. For studies that did not include information that we needed for this systematic review and meta-analysis, we contacted the authors requesting for more information pertaining to their study. When data were not clearly reported after contacting the authors or no reply was received from the authors, these data were coded as ′not reported (NR)′.

### 4.3. Data Extraction and Quality Assessment

All eligible articles were evaluated independently by two raters (P.P. and S.C.). Inter-rater discrepancies were reconciled by consensus or by a third arbitrator (P.S.). We extracted the following variables from selected studies with a pre-designed sheet: first author; country of the study; year of publication; IP-10 assay; clinical specimens; unit of measurement for the IP-10; study population type; the criteria for determining disease activity, such as disease activity index or renal pathology grade; definition of active disease; value of IP-10 in each subgroup of patients; correlation coefficient between IP-10 and disease activity; correlation coefficient between IP-10 and other conventional biomarkers, such as complement level and anti-dsDNA antibody. The methodological quality of the selected studies was assessed by two independent raters (P.P. and S.C.) using the Quality Assessment Tool for Diagnostic Accuracy Studies- (QUADAS-) 2 tool [43]. The QUADAS-2 tool provided the systematic quality assessment of the diagnostic studies for systematic review by pointing out the risk of bias and applicability concerns among included studies according to four key domains: patient selection; index test; reference standard; flow and timing. Disagreement on the assessment of quality appraisal items were reconciled by consensus or by a third rater (P.S.).

### 4.4. Data Synthesis and Statistical Analysis

In the studies that provided quantitative values of serum or urine IP-10 level, and/or their correlation coefficient with the disease activity index or other conventional biomarkers, the data were used in the meta-analysis. However, all relevant results from each study, including those not eligible for meta-analysis, were also summarized. Comparison of serum and urine IP-10 levels between groups was performed by using the mean difference (MD). Correlation between serum and urine IP-10, and disease activity by various clinical activity scoring or other biomarkers were demonstrated using pooled correlation analysis. Heterogeneity was assessed using the *I*^2^ index and Q test *p*-value. Data were analyzed using a Comprehensive Meta-Analysis software [44]. A two-sided *p*-value of less than 0.05 was considered statistically significant. The risk of publication bias was also evaluated by Egger’s test [45].

## 5. Conclusions

This systemic review and meta-analysis is the first of its kind to combine all available data of IP-10 and confirm its merit. Serum IP-10 is significantly elevated in SLE patients and positively correlated with overall disease activity. Urine IP-10, on the other hand, can be a useful biomarker for detecting active LN but it still needs a standardized method of measurement, as well as prospective studies correlating with biopsy-proven active LN to establish its value.

## Figures and Tables

**Figure 1 ijms-20-04954-f001:**
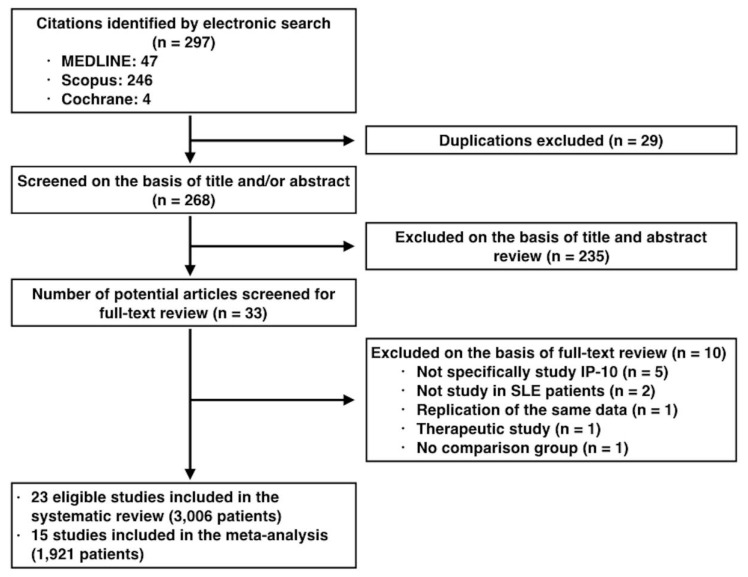
Study selection flow diagram.

**Figure 2 ijms-20-04954-f002:**
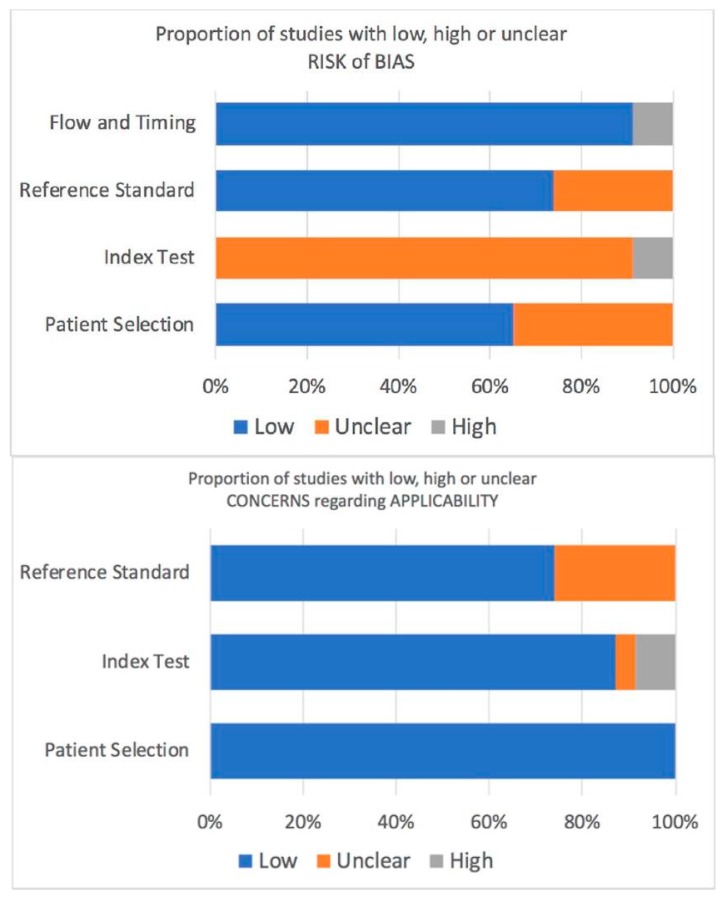
Risk of bias and applicability concerns graph according to the QUADAS-2 tool.

**Table 1 ijms-20-04954-t001:** Characteristics of the studies included in the systematic review.

Study	Country	N	SLE Patients	LN Patients	Disease Activity Criteria	Specimen	IP-10 Assay
Narumi (2000) [17]	Japan	65	28	10	NR	Serum	ELISA (in-house)
Eriksson (2003) [21]	Sweden	45	23	10	SLEDAI ≥ 6	Serum	ELISA (R&D)
Avihingsanon (2006) [13]	Thailand	36	26	26	ISN/RPS class IV LN	Urine	qRT-PCR
Lit (2006) [14]	Hong Kong	120	80	26	SLEDAI ≥ 6	Serum	CBA
Bauer (2009) [15]	US	267	267	NR	SLEDAI ≥ 6	Serum	ELISA (SearchLight)
Kong (2009) [16]	Singapore	514	464	NR	SLAM-R > 5 *	Serum	ELISA (OptEIA)
Morimoto (2009) [22]	Japan	71	41	41	SLEDAI > 8	Serum	ELISA (BD BioScience)
Wong (2009) [23]	Hong Kong	37	23	NR	NR	Serum	CBA
Shah (2011) [24]	India	60	30	NR	SLEDAI **	Serum	ELISA (OptEIA)
Bjorkander (2012) [20]	Sweden	35	15	3	SLEDAI ≥ 4	Serum	CBA
Watson (2012) [19]	UK	83	60	8	BILAG A or B	Urine	ELISA (R&D)
Abujam (2013) [25]	India	136	136	46	SELENA-SLEDAI ≥ 4, active LN if proteinuria ≥ 500 mg/day or active urinary sediment	Serum & urine	ELISA (OptEIA)
Hrycek (2013) [26]	Poland	77	48	NR	NR	Serum	ELISA (RAYBIO)
Marie (2013) [27]	Egypt	60	60	30	NR	Urine	ELISA (OptEIA)
Rose (2013) [28]	Germany	106	79	17	BILAG (not clearly stated)	Serum	ELISA (R&D)
Doe (2014) [29]	Japan	189	80	NR	NR	Serum	ELISA (R&D)
Dominguez-Gutierrez (2014) [30]	US	168	103	NR	SLEDAI > 4	Serum	qRT-PCR
El-Gohary (2016) [31]	Egypt	40	30	6	SLEDAI ≥ 4, active LN if proteinuria ≥ 500 mg/day or active urinary sediment	Serum & urine	ELISA (R&D)
Choe (2016) [32]	Korea	131	70	32	SLEDAI ≥ 6	Serum & urine	ELISA (R&D)
Odler (2017) [33]	Hungary	27	18	NR	NR	Serum	ELISA (custom made array kits)
Oke (2017) [34]	Sweden	522	261	NR	SLEDAI > 6	Serum	ELISA (R&D)
Wu (2017) [35]	China	191	111	NR	NR	Serum	Multiplex immunoassay (Bio-Plex 200)
Rose (2017) [18]	Germany	26	26	NR	BILAG index classifying patients into flare, inactive, and remitting status	Serum	ELISA (R&D)

* SLAM-R ≤ 5 was considered as having inactive disease while SLAM-R of >5 to ≤10, >10 to ≤15, and >15 were considered as having mild, moderate, and severe disease activity, respectively. ** SLEDAI score <10, 10–20, >20 were considered as having a mild, moderate, or severe disease activity, respectively.

**Table 2 ijms-20-04954-t002:** Mean difference between each subgroup comparison.

Study Outcomes	Number of Studies/Study Arms	Number of Patients	Mean Difference (95% CI)	*p*-Value	*I*^2^ Index ^a^	*p*-Value	Egger′s Test ^b^ *p*-Value
Serum IP-10 in SLE (pg/mL)							
• Serum IP-10 in SLE vs healthy controls	8/8	1069	153.86 (91.63 to 216.10)	<0.001	87.1	<0.001	0.04
• Serum IP-10 in active SLE vs inactive SLE	4/5	879	356.51 (59.57 to 653.44)	0.019	95.9	<0.001	0.28
Serum IP-10 in LN (pg/mL)							
• Serum IP-10 in LN vs healthy controls	3/3	193	183.84 (126.54 to 241.14)	<0.001	0	0.37	0.49
• Serum IP-10 in LN vs SLE without LN ^c^	5/7	402	22.62 (−182.83 to 228.08)	0.83	86.9	<0.001	0.22
Urine IP-10 in SLE (pg/mgCr x 100)							
• Urine IP-10 in SLE vs healthy controls	2/2	95	0.21 (−0.74 to 1.15)	0.67	54.3	0.14	NA
• Urine IP-10 in SLE vs inactive SLE	2/2	156	2.81 (−2.40 to 8.01)	0.29	90.1	0.001	NA
Urine IP-10 in LN (pg/mgCr x 100)							
• Urine IP-10 in LN vs healthy controls	2/2	51	0.21 (−0.92 to 1.33)	0.72	17.2	0.27	NA
• Urine IP-10 in LN vs SLE without LN ^c^	3/5	282	3.47 (−0.18 to 7.12)	0.06	88.2	<0.001	0.13

^a^ Test for study heterogeneity; ^b^ test for publication bias.; ^c^ include inactive SLE and active SLE without renal involvement; abbreviations: BILAG, British Isles Lupus Assessment Group; ESR, erythrocyte sedimentation rate; IP-10, interferon-inducible protein 10; LN, lupus nephritis; NA, not available; SLE, systemic lupus erythematosus; SLEDAI, SLE disease activity index.

**Table 3 ijms-20-04954-t003:** Pooled correlation coefficient between serum or urine IP-10 and SLE/LN disease activity or other biomarkers.

Study Outcomes	Number of Studies/Study Arms	Number of Patients	Pooled Correlation Coefficient (95% CI)	*p*-Value	*I*^2^ index ^a^	*p*-Value	Egger’s Test ^b^ *p*-Value
Serum IP-10 and disease activity							
• Serum IP-10 and SLEDAI	7/7	802	0.29 (0.22 to 0.35)	<0.001	0	0.61	0.48
• Serum IP-10 and BILAG index	2/2	105	0.41 (0.24 to 0.56)	<0.001	0	0.55	NA
Urine IP-10 and disease activity							
• Urine IP-10 and SLEDAI	3/3	236	0.21 (0.05 to 0.36)	0.011	27.4	0.25	0.53
• Urine IP-10 and renal SLEDAI	3/3	236	0.29 (0.05 to 0.50)	0.019	73.8	0.02	0.85
Serum IP-10 and other biomarkers							
• Serum IP-10 and complement level	3/5	1096	−0.20 (−0.30 to −0.10)	<0.001	51.3	0.08	0.08
• Serum IP-10 and anti-dsDNA and ESR	3/5	1096	0.28 (0.15 to 0.40)	<0.001	70.3	0.01	0.07

^a^ Test for study heterogeneity; ^b^ test for publication bias; abbreviations: BILAG, British Isles Lupus Assessment Group; ESR, erythrocyte sedimentation rate; IP-10, interferon-inducible protein 10; LN, lupus nephritis; NA, not available; SLE, systemic lupus erythematosus; SLEDAI, SLE disease activity index.

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
