# Peer review of "Interferon-Inducible Protein 10 and Disease Activity in Systemic Lupus Erythematosus and Lupus Nephritis: A Systematic Review and Meta-Analysis"

_ijms, 2019, doi:10.3390/ijms20194954_

Round 1

Reviewer 1 Report

Considering the important pathogenic role of IFNs demonstrated in human lupus patients and experimental SLE models, it is necessary that we know as much as we can about IFN-inducible proteins and their effects in SLESLE. Interestingly, IFN itself is difficult to assess but ISGs ISGs are easy to detect and are an important signature of IFN SLE activity. With the current interest of the lupus community in targeting IFNs for therapeutics, this study provides a detailed analysis of one of the important ISG. 

The significance of the study is undeniable. The authors have done well to objectively describe all the reports. The meta-data analysis is comprehensive. As noted by the authors at several places, the variety in data collection and analysis introduces variables in each study which are hard to put together during a meta-analysis. The challenges of such a study are clearly put forward. 

The authors should however discuss some more on the importance or significance of this exercise. The short introduction does not sufficiently justify why IP-10 was chosen for this analysis and how the meta-analysis helps and what could be the future of studying IP-10, more than just as a biomarker of LN or SLE. 

An interesting positive correlation, albeit nominal, is seen between IP-10 and dsDNA. dsDNA is positively associated with lupus flares and also LN. Additionally, the role of T-cells in dsDNA production is crucial. It will add value to the manuscript if the authors are able to connect some of the conclusive data together and present a probable holistic picture as food for thought for the readers. 

Author Response

Points 1: The authors should however discuss some more on the importance or significance of this exercise. The short introduction does not sufficiently justify why IP-10 was chosen for this analysis and how the meta-analysis helps and what could be the future of studying IP-10, more than just as a biomarker of LN or SLE.

Response 1:

Thank you for your kind suggestion. We have added biological background of the interferon-related pathway and IP-10 to the introduction as follows: “It has been established that both innate and adaptive immune response were involved in SLE pathogenesis, especially interferon (IFN) signaling pathway (Ytterberg et al. Arthritis & Rheumatism. 1982;25(4):401-6, Hooks et al. New England Journal of Medicine. 1979;301(1):5-8). More recently, gene expression studies exhibited an overexpression of IFN-stimulated genes (ISGs) in blood of SLE patients highlighting the significant role of IFN in SLE (Bennett et al. J Exp Med. 2003;197(6):711-23, Baechler et al. Proc Natl Acad Sci U S A. 2003;100(5):2610-5). Further study on ISG products identifies many promising IFN-regulated chemokines. IFN gamma inducible protein 10 (IP-10) or chemokine ligand 10 (CXCL10) is of particular interest being one of the IFN-regulated chemokines that show the strongest correlation with SLE disease activity from the earlier study (Bauer et al. PLoS Med. 2006;3(12):e491).

Point 2: An interesting positive correlation, albeit nominal, is seen between IP-10 and dsDNA. dsDNA is positively associated with lupus flares and also LN. Additionally, the role of T-cells in dsDNA production is crucial. It will add value to the manuscript if the authors are able to connect some of the conclusive data together and present a probable holistic picture as food for thought for the readers.

Response 2:

Correlation between IP-10 and anti-dsDNA antibody is, indeed, very interesting and warrants further discussion. Three studies demonstrated positive correlation between serum IP-10 and anti-dsDNA antibody, among SLE patients (Kong et al. Clinical and Experimental Immunology. 2009;156(1):134-40, Narumi et al. Cytokine. 2000;12(10):1561-5, Choe et al. Inflammation Research 2016;65(6):479-88). One study even showed significant correlation during serial measurements. This significant correlation confirms their connection in the pathogenesis of SLE. Basically, anti-dsDNA antibody can form immune complexes and activate IFN signaling pathway which ultimately leads to IP-10 production. Anyhow, anti-dsDNA antibody is still far from being an ideal biomarker for predicting SLE disease activity. Persistent elevation of anti-dsDNA antibody with no evidence of active disease or obvious disease activity with normal anti-dsDNA antibody levels have been reported (Reveille JD. Lupus. 2004;13(5):290-7). This infers that the inducer of IFN signaling pathway in SLE is not limited to anti-dsDNA antibody. IP-10, on the other hand, represents the downstream mediators of the IFN-regulated pathways and may better correlates with disease activity. However, studies comparing area under the ROC curve analysis between IP-10 and anti-dsDNA antibody are limited and yielded conflicting results (Kong et al. Clinical and Experimental Immunology. 2009;156(1):134-40, Rose et al. Rheumatology 2017;56(9):1618-26, Abujam et al. Lupus. 2013;22(6):614-23.

Reviewer 2 Report

  It is an interesting review article with meta-analysis examining the interferon-inducible protein 10 (IP-10) and disease activity in systemic lupus erythematosus (SLE) and lupus nephritis (LN). The manuscript is well written in English and the findings are relevant to the potential clinical application. Only one minor point needs to be clarified as follows.

  Although the authors concluded that serum IP-10 was correlated with SLE disease activity whereas urine IP-10 was a biomarker for detecting active LN. Most cited studies were done by using blood or urine samples alone. The authors should put emphasis on the references 21 (Lupus 2013;22:614) and 28 (Inflammation research 2016;65:479) which were carried out by simultaneously analyzing both serum and urine IP-10 concentrations, and further discuss this issue.

Author Response

Points 1: Although the authors concluded that serum IP-10 was correlated with SLE disease activity whereas urine IP-10 was a biomarker for detecting active LN. Most cited studies were done by using blood or urine samples alone. The authors should put emphasis on the references 21 (Lupus 2013;22:614) and 28 (Inflammation research 2016;65:479) which were carried out by simultaneously analyzing both serum and urine IP-10 concentrations, and further discuss this issue.

Response 1:

Thank you for your kind suggestion. We have added more information to the results section as follows: Three studies measured both serum and urine IP-10 in each patients (n = 307 patients, 80 active non-LN patients, 84 active LN patients, 72 inactive SLE patients, and 71 healthy controls) (El-gohary et al. J Immunol Res 2016, Abujam et al. Lupus 2013, Choe et al. Inflammation Research 2016). Although the data could not be meta-analyzed, all three studies provided area under the ROC curve analysis. Only a study by El-gohary et al analyzed both serum and urine IP-10 and showed that serum IP-10 was better than urine IP-10 in differentiating active from inactive SLE (area under the ROC curve 0.753, 95% CI 0.594-0.911 vs 0.654, 95% CI 0.467-0.841, respectively). Another study by Abujam et al only analyzed serum IP-10 for SLE and urine IP-10 for LN and hence did not provide direct comparison of performance between serum and urine IP-10. The other study by Choe et al focused mainly on LN and found that serum and urine IP-10 performed comparably in detecting SLE with LN (area under the ROC curve 0.595, 95% CI 0.46-0.73 vs 0.519, 95% CI 0.38-0.66, respectively).